# CAN GRPO HELP LLMS TRANSCEND THEIR PRETRAINING ORIGIN?

## ABSTRACT

Reinforcement Learning with Verifiable Rewards (RLVR), primarily driven by the Group Relative Policy Optimization (GRPO) algorithm, is a leading approach for enhancing the reasoning abilities of Large Language Models (LLMs). Despite its wide adoption, GRPO's gains are often inconsistent; for instance, a model may show significant improvement in one reasoning domain, like mathematics, yet remain stagnant in another, such as medicine. This inconsistency raises a critical question: under what conditions does GRPO improve reasoning and generalize out-of-distribution (OOD)? We investigate this from a data distribution perspective. We first prove theoretically that GRPO is a conservative reweighting scheme, bounded by the base model's distribution and thus unable to discover completely novel solutions. We further validate this in carefully designed controlled studies by training transformers from scratch, evaluating generalization across reasoning depth, input length, token representation, and compositionality. Our results provide a principled explanation for GRPO's boundaries: OOD improvement emerges only when the target task aligns with the model's pretrained biases, while gains on in-distribution (ID) tasks diminish as performance saturates. This reframes GRPO not as a universal reasoning enhancer but as a tool that sharpens pretraining biases. Our findings motivate future development of algorithms that can expand a model's capabilities beyond its pretraining origin.

## 1 INTRODUCTION

Reinforcement Learning with Verifiable Rewards (RLVR), powered by the Group Relative Policy Optimization (GRPO) algorithm and its variants (Shao et al., 2024), has become a dominant approach to advancing the reasoning capabilities of Large Language Models (LLMs). This approach is credited with state-of-the-art performance in complex reasoning domains, such as math problem-solving and programming (Wang et al., 2025c; Su et al., 2025). However, a critical paradox casts doubts on its success: substantial performance gains from GRPO can be achieved even with spurious rewards on math reasoning tasks (Shao et al., 2025). In particular, our preliminary studies highlight that GRPO's effectiveness varies significantly across models and domains, as shown in Figure 1. Specifically, Qwen models achieve larger gains on math and general-domain datasets, whereas Llama models achieve larger gains on general and clinical-domain datasets. The hyperparameters are detailed in Appendix B.1. This phenomenon raises a fundamental question:

> ***Under what conditions does GRPO enhance the reasoning capabilities of an LLM?***

The mechanism behind RLVR is a subject of ongoing discussion. One line of research frames RLVR as a conservative variant of Supervised Fine-Tuning (SFT), showing its limited capacity to generalize out-of-distribution (OOD) (Samineni et al., 2025; Wu et al., 2025c). Others suggest that RLVR drives models toward dominant output distributions inherited from pretraining data (Zhao et al., 2025b) and amplifies solutions already present in the base model—raising pass@1 by improving sampling efficiency while reducing coverage at larger pass@k (Yue et al., 2025; Wu et al., 2025a). Nevertheless, prior studies have predominantly relied on pretrained, off-the-shelf LLMs and web-scale datasets online. This setting makes it difficult to disentangle the true effects of RLVR from confounding variables such as the composition of the pretraining data and undisclosed training details. Furthermore, the heterogeneity of these online datasets makes it exceptionally challenging to isolate different OOD settings, preventing a systematic evaluation of generalization.

Our work instead focuses on analyzing the effects of pretraining data distribution on GRPO through a controlled study that isolates fundamental OOD scenarios. We ***hypothesize*** that GRPO's effectiveness is determined by the distribution alignment between the model's pretrained inductive biases

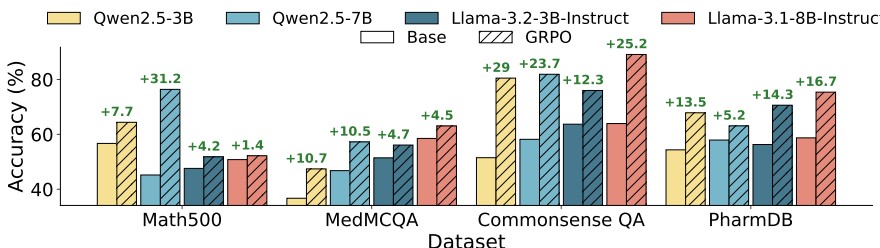

Figure 1: Accuracy on various datasets after applying GRPO on LLMs. We benchmark GRPO on four LLMs (Qwen2.5-3B, Qwen2.5-7B (Qwen et al., 2025), Llama3.2-3B-Instruct, Llama-3.1-8B-Instruct (Grattafiori et al., 2024)) across four datasets (Math500 (Hendrycks et al., 2021), MedM-CQA (Pal et al., 2022), PharmDB (Abdullahi et al., 2025), CommonsenseQA (Talmor et al., 2019)). The results show that Qwen models achieve larger gains on math and general-domain datasets, whereas Llama models achieves larger gains on general and clinical-domain datasets. Notably, Qwen2.5-7B improves by 31.2% on Math500 but only 5.2% on PharmDB; Llama-3.1-8B-Instruct gains 16.7% on PharmDB but merely 1.4% on Math500. These findings highlight that GRPO's effectiveness is highly dependent on the specific model–dataset pairing.

and the target task. To eliminate confounding variables, we train transformer models from scratch on carefully synthesized data, granting us full control over data distributions and models. Therefore, we can directly observe the causal effects of pretraining data composition on GRPO's effectiveness.

We begin by presenting a theoretical analysis that characterizes GRPO as a conservative reweighting scheme bounded by the base model's policy, incapable of discovering completely novel solutions. Guided by this theoretical foundation, we design a series of controlled experiments with synthetic data to test the generalization limits of GRPO. We evaluate its performance across four distinct OOD scenarios: **(1) reasoning depth**, by varying the number of required reasoning steps; **(2) input length**, by altering sequence lengths; **(3) token representation**, by representing familiar concepts with novel tokens; and **(4) compositional reasoning**, by requiring the model to combine known skills in new combinations. Our empirical findings consistently validate our hypothesis: GRPO only facilitates OOD generalization when there is a significant overlap between in-distribution (ID) and OOD tasks. Increasing the proportion of OOD data during pretraining improves the alignment between the base model and the task, resulting in larger performance gains, although these gains diminish as performance saturates. This reframes GRPO not as a universal reasoning enhancer but as a tool that sharpens existing biases. This offers important implications for its practical application and for future research on its generalization boundaries. Our contributions are as follows:

- **Real-world Motivation**: We show that GRPO yields inconsistent gains across a range of LLMs and datasets, motivating the need for a systematic investigation into its underlying mechanisms.

- **Theoretical Analysis**: We characterize GRPO as a conservative reweighting mechanism, proving that its gains depend on the base model's initial capabilities for correctly answering the prompts, and thus show that GRPO cannot discover novel reasoning patterns.

- **Controlled Experimental Design**: We introduce a principled methodology, training transformers from scratch on synthetic data to probe GRPO's generalization limits across four axes: reasoning depth, input length, token representation, and compositional reasoning.

- **Empirical Findings**: Across four controlled settings, we show that GRPO's OOD generalization is contingent on the alignment between the model's pretrained distribution and the target task, solidifying that it sharpens pre-existing biases rather than generating novel reasoning.

## 2 RELATED WORKS

**RLVR.** Reinforcement Learning with Verifiable Rewards (RLVR) has become the standard for improving the reasoning capabilities of Large Language Models (LLMs). The foundational algorithm in this area is Group Relative Policy Optimization (GRPO) (Shao et al., 2024), which uses outcome-based rewards and group-wise advantage computations to obviate the need for a reward model. Subsequently, variants of GRPO are proposed. Dr. GRPO (Liu et al., 2025) removes the standard deviation normalization term in advantage computation to reduce generating longer incorrect responses, while Dynamic Sampling Policy Optimization (DAPO) (Yu et al., 2025) introduces a "Clip-Higher" strategy to encourage exploration and uses token-level loss averaging for more fine-grained loss

attribution. Concurrently, Dynamic Clipping Policy Optimization (DCPO) (Yang et al., 2025) addresses the zero-gradient problem with dynamic, probability-aware clipping and reduces variance via advantage standardization. To understand the mechanisms driving the success of RLVR, we ground our analysis in the original GRPO. Analyzing this foundational algorithm allows us to disentangle the core mechanisms from other heuristics introduced by its variants and ensures that our insights are applicable across all of them.

**RLVR Generalization.** While RLVR shows promise for enhancing the reasoning capabilities of LLMs, its effectiveness has proven highly inconsistent across models and datasets (Wang et al., 2025a; Sun et al., 2025; Shao et al., 2025). To explain these phenomena, some work concludes that RLVR is a conservative variant of SFT, making it inherently less effective at promoting out-of-distribution (OOD) generalization (Samineni et al., 2025; Wu et al., 2025c; Lv et al., 2025). Others argue that RLVR's benefits stem from improving sampling efficiency of existing solutions and patterns in the base model (Wu et al., 2025b; Zhao et al., 2025b; Yue et al., 2025; Wu et al., 2025a). However, these studies focus on off-the-shelf LLMs and web-scale datasets. This introduces confounding factors—such as unknown pretraining procedures and heterogeneous data composition—that prevent a systematic evaluation. In response, our work adopts a controlled experimental setup that evaluates fundamental OOD scenarios. We pretrain transformer models from scratch on synthetic data to enable comprehensive analysis of generalization limits across four axes, including reasoning depth, input length, token representation, and compositionality.

**OOD Generalization for LLMs.** Generalizing to OOD data remains an open challenge for LLMs. In particular, models tend to fall back on patterns from their pretraining data rather than learning genuinely new rules through in-context learning (ICL), even when their successes stem from learned compositional abilities (Wang et al., 2024; 2025b; Song et al., 2024). While techniques like Chain-of-Thought (CoT) (Wei et al., 2022) can improve reasoning, their effectiveness depends heavily on the alignment between supervision, inductive bias, and task structure (Yao et al., 2025; Cho et al.). This robustness is often fragile, collapsing into mere pattern recognition when tested rigorously (Zhang et al., 2024; Wang et al., 2025d; Zhao et al., 2025a). This general tendency for LLMs to favor pattern matching over true generalization provides the context for our study. RLVR is designed to improve reasoning, but if it cannot overcome this fundamental OOD barrier, its benefits are limited. We therefore analyze GRPO to investigate the fundamental generalization limits of RLVR.

## 3 GRPO AS A CONSERVATIVE REWEIGHTING MECHANISM

The behavior of GRPO can be understood by analyzing the RL optimization objective. The theoretical optimal policy has an exponential-tilting form, which simply reweights the base distribution. This reweighting preserves the support of the base model while amplifying probability mass on responses that receive a reward. As a result, the effect of GRPO is determined by how much probability the pretrained model already assigns to correct answers. In the following, we formalize this dependence and show that GRPO strengthens existing overlap with correct solutions but cannot discover solutions absent from the base distribution.

### 3.1 THE DEPENDENCE OF THE OPTIMAL POLICY ON PRETRAINED MASS

**Preliminaries.** Consider a prompt $x$ from a distribution $d$, with a space of possible token sequence responses $\mathcal{Y}$. We have a base policy $q(\cdot \mid x)$ and a binary reward function $R(x, y) \in \{0, 1\}$ that verifies if a response $y$ is correct. GRPO and other similar RLVR methods all optimize variants of the following objective:

$$\mathcal{J}_\beta(\pi) = \mathbb{E}_{x \sim d}\Big[\mathbb{E}_{y \sim \pi(\cdot|x)} R(x, y)\Big] - \beta\, \mathbb{E}_{x \sim d}\Big[\mathrm{KL}\big(\pi(\cdot \mid x) \,\|\, q(\cdot \mid x)\big)\Big], \qquad \beta > 0. \tag{1}$$

The solution to this optimization problem has an exponential-tilting form:

$$\pi_\beta^\star(y \mid x) \;\propto\; q(y \mid x) \exp\big(\beta^{-1} R(x, y)\big) \quad \text{for each } x \in \mathcal{X}. \tag{2}$$

The optimal policy $\pi_\beta^\star$ is the base policy $q$ re-weighted. The probability of correct responses (where $R(x, y) = 1$) is amplified by a factor of $e^{\beta^{-1}}$. To formalize this, define the set of all correct responses for a given prompt $x$ as $C_x = \{y \in \mathcal{Y} : R(x, y) = 1\}$. We can then define the pretrained correct mass as the total probability the pretrained model assigns to all correct answers:

$$Q(x) \coloneqq q(C_x \mid x). \tag{3}$$

$Q(x)$ captures the extent of the model's pretrained capacity for prompt $x$. We now express both the finetuned performance and its *marginal gain over the base model* in terms of this value.

**Theorem 1 (Pointwise performance depends only on pretrained overlap).** *Under this objective with binary rewards, the probability of generating a correct response for any prompt $x$ is*

$$\pi_\beta^\star(C_x \mid x) = \frac{Q(x)a}{(1 - Q(x)) + Q(x)a} =: f_\beta(Q(x)). \quad \left(a = e^{\beta^{-1}}\right) \tag{4}$$

*Moreover, for any fixed $\beta > 0$: (i) $f_\beta$ is strictly increasing, (ii) if $Q(x) = 0$ then $\pi_\beta^\star(C_x \mid x) = 0$, and (iii) improvement saturates as $Q(x) \to 1$ (Proof in Appendix C.1).*

Define the *marginal gain* (performance increment over the base model) for a prompt $x$ as

$$\mathrm{MGain}_\beta(x) := g_\beta(Q(x)) = f_\beta(Q(x)) - Q(x).$$

Part (ii) implies a hard limit: if the pretrained model assigns zero probability to correct answers, then no positive marginal gain is possible.

**Corollary 1 (Marginal gain monotonicity.** *Suppose a base model $q'$ is better aligned than $q$ (i.e., $Q'(x) \geq Q(x)$ for all $x$). If for all $x$ the overlap satisfies*

$$Q(x) \leq \frac{\sqrt{a} - 1}{a - 1} \quad \left(a = e^{\beta^{-1}}\right), \quad \mathbb{E}_{x \sim d}\left[g_\beta(Q'(x))\right] \geq \mathbb{E}_{x \sim d}\left[g_\beta(Q(x))\right].$$

*That is, there exists a certain point such that below that point, better model alignment guarantees larger marginal improvement from finetuning. When $Q(x)$ is large, the marginal gain diminishes due to saturation. Since $f_\beta(Q) \to 1$ as $Q \to 1$, we have $g_\beta(Q) \to 0$ (Proof in Appendix C.2).*

### 3.2 PERFORMANCE LIMITS IN THE LOW-MASS REGIME

In practice, while LLMs rarely assign exactly zero probability to any token, the probability of a specific sequence can be astronomically small. Hence the correct mass $Q(x)$ is often tiny (though nonzero). The next propositions quantify what this means for finetuning.

**Proposition 1 (Small-mass regime is first-order in $Q$).** *For every $x$ and $\beta > 0$, the marginal gain is bounded linearly by $Q(x)$:*

$$\pi_\beta^\star(C_x \mid x) - Q(x) \leq \left(e^{\beta^{-1}} - 1\right) Q(x). \tag{5}$$

When $Q(x) \ll 1$, finetuning can only scale the base model's already tiny correct mass; the improvement is at most a constant factor times $Q(x)$ (Proof in Appendix C.3).

**Proposition 2 (Sequence-level gains can be exponentially small).** *Even if every token has a minimum probability $q_{\mathrm{tok}}(a \mid \cdot) \geq \eta > 0$, the total correct mass for a sequence of length $T$ can decay exponentially as $Q(x) \leq |C_x|\eta^T$. Consequently, achieving a non-negligible gain requires the reward amplification to grow with sequence length, i.e., $\beta^{-1} = \Omega\left(T \log(1/\eta)\right)$.*

Refer to proof in Appendix C.4. These results reveal a low-mass bottleneck. Proposition 2 shows that for longer sequences, the initial correct mass $Q(x)$ can become exponentially small. Since Proposition 1 establishes that finetuning gains scale at most linearly with this tiny mass, GRPO's effectiveness is severely limited. In short, GRPO amplifies what the pretrained model already knows, but cannot discover solutions from scratch when their initial probability is negligible.

## 4 LEVERAGE SYNTHETIC DATA TO PROBE THE GENERALIZATION OF GRPO

To evaluate GRPO's generalization capabilities, we use a controlled framework with four synthetic datasets. Each is designed to isolate a distinct distributional shift (e.g., reasoning depth, input length, token representation, or compositional reasoning) and is split into in-distribution ($\mathcal{D}_{\mathrm{ID}}$) and out-of-distribution ($\mathcal{D}_{\mathrm{OOD}}$) sources, as illustrated in Figure 2 and Table 1. For each of the four settings, we train transformer models from scratch using a three-stage pipeline: pretraining on 67 million samples, followed by fine-tuning with 2,000 SFT examples and 1,000 GRPO samples (see Appendix B.2 for full hyperparameter details). Pre-training uses a mixture of $\mathcal{D}_{\mathrm{ID}}$ and $\mathcal{D}_{\mathrm{OOD}}$, while SFT use only $\mathcal{D}_{\mathrm{ID}}$ and GRPO uses either $\mathcal{D}_{\mathrm{ID}}$ or $\mathcal{D}_{\mathrm{OOD}}$. This totals 268 million unique samples across all experiments. We measure performance on ID test sets (to assess retention) against OOD test sets (to assess generalization), revealing how GRPO handles specific generalization challenges.

Figure 2: Our overall framework to probe the generalization limits of GRPO. We train transformers from scratch with synthetic data and evaluate generalization in four settings: (1) reasoning depth, (2) input length, (3) token representation, and (4) compositional reasoning.

## 4.1 TASK DEFINITON

To construct tasks for LLMs to be trained on, we consider operations over fixed-length strings. Let $\Sigma$ be a finite alphabet and $k \geq 1$. For an input $x = (x_0, x_1, \ldots, x_{k-1}) \in \Sigma^k$, we define two operations: *graph traversal* and *cyclic shift*.

**Graph Traversal.** Let $\sigma : \Sigma \to \Sigma$ be a permutation, represented as a mapping of a directed graph $G_\sigma = (\Sigma, \{(u, \sigma(u)) : u \in \Sigma\})$. The traversal operator $T_\sigma$ applies $\sigma$ to each symbol:

$$T_\sigma(x_0, \ldots, x_{k-1}) = (\sigma(x_0), \ldots, \sigma(x_{k-1})).$$

**Cyclic Shift.** The shift operator $R$ performs a one-step left rotation of the sequence:

$$R(x_0, x_1, \ldots, x_{k-1}) = (x_1, x_2, \ldots, x_{k-1}, x_0).$$

**Composition.** Let $S = (f_1, f_2, \ldots, f_m)$ be a sequence of operators, where each $f_i \in \{T, R\}$. Then the composition $f_S = f_m \circ f_{m-1} \circ \cdots \circ f_1$, and the output for input $x$ is $\hat{x} = f_S(x)$.

**Learning Task.** The model is trained to approximate the mapping $x \mapsto \hat{x}$, given an input sequence $x$ and an operator sequence $S$. Each intermediate state $e^{(j)} = f_j \circ \cdots \circ f_1(x)$ serves as a reasoning step to reach the output.

| Test | Label | Input | Output |
|---|---|---|---|
| Reasoning Depth | 1 Step | TSKE3 `<trav>` | $\Rightarrow$ 4EUOT |
| | 2 Steps | TSKE3 `<trav><trav>` | $\Rightarrow$ 4EUOT `<trav>` $\Rightarrow$ RO1K4 |
| | 3 Steps | TSKE3 `<trav><trav><trav>` | $\Rightarrow$ 4EUOT `<trav><trav>` $\Rightarrow$ RO1K4 `<trav>` $\Rightarrow$ VKDUR |
| Input Length | 5 Chars | 4CMKQ+++ `<trav>` | $\Rightarrow$ RG6U5+++ |
| | 6 Chars | 4CMKQE++ `<trav>` | $\Rightarrow$ RG6U5O++ |
| | 7 Chars | 4CMKQE6+ `<trav>` | $\Rightarrow$ RG6U5OJ+ |
| Token Rep. | Orig. | EOCNS `<trav>` | $\Rightarrow$ RGUSP |
| | Alt. | eocns `<trav>` | $\Rightarrow$ rgusp |
| | Mixed | EoCNs `<trav>` | $\Rightarrow$ RgUSp |
| Composition | Trav-Only | TSKE3 `<trav><trav>` | $\Rightarrow$ 4EUOT `<trav>` $\Rightarrow$ RO1K4 |
| | Shift-Only | TSKE3 `<shift><shift>` | $\Rightarrow$ SKE3T `<shift>` $\Rightarrow$ KE3TS |
| | Composite | TSKE3 `<trav><shift>` | $\Rightarrow$ 4EUOT `<shift>` $\Rightarrow$ EUOT4 |

Table 1: Synthetic data samples for each generalization test.

## 4.2 ID–OOD DEFINITION

Let a dataset instance be a triple $(x, S, \hat{x})$, where $x \in \Sigma^k$ is an input of length $k$ from an alphabet $\Sigma$, $S = (f_1, \ldots, f_m)$ is a sequence of operators from $\{T_\sigma, R\}$, and $\hat{x} = f_S(x)$ is the output.

- **Reasoning Depth.** Fix the token $\Sigma$, graph mapping $\sigma$, and input length $k = 5$, so any input $x \in \Sigma^5$. Let $\mathcal{G}_m$ be the set of operators where $S$ is $m$ steps of $T_\sigma$. We change the number of reasoning steps ($m$) to construct ID and OOD datasets:

$$\mathcal{D}_{\text{ID}}^{\text{depth}} = \{(x, S, \hat{x}) : S \in \mathcal{G}_1 \cup \mathcal{G}_2\}, \quad \mathcal{D}_{\text{OOD}}^{\text{depth}} = \{(x, S, \hat{x}) : S \in \mathcal{G}_3\}.$$

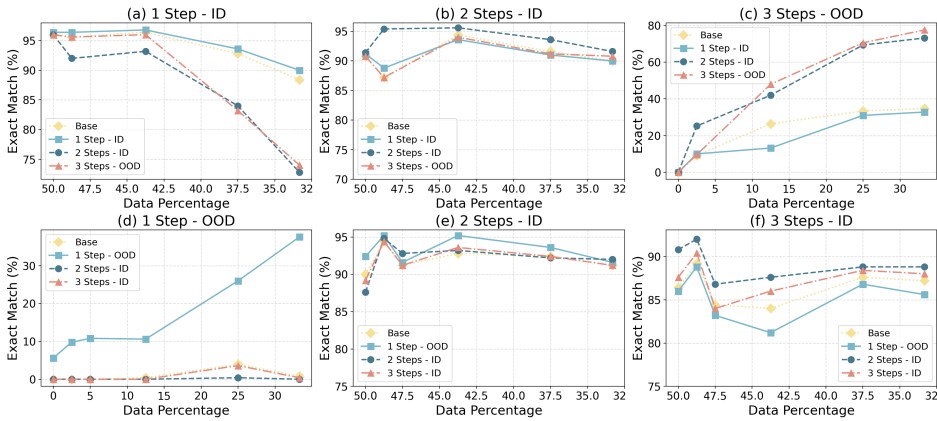

Figure 3: Reasoning depth generalization. Top row (a–c): models pretrained on shallow reasoning (1–2 steps, ID) mixed with deeper reasoning (3 steps, OOD). Bottom row (d–f): models pretrained on deep reasoning (2–3 steps, ID) mixed with shallower reasoning (1 step, OOD).

- **Input Length.** Fix the token $\Sigma$ and graph mapping $\sigma$, Set the operator $S = (T_\sigma)$. We vary the input length $(k)$ to construct ID and OOD datasets:

$$\mathcal{D}_{\text{ID}}^{\text{len}} = \{(x, S, \hat{x}) : x \in \Sigma^k, \ k \in \{5, 6\}\}, \quad \mathcal{D}_{\text{OOD}}^{\text{len}} = \{(x, S, \hat{x}) : x \in \Sigma^7\}.$$

- **Token Representation.** Fix the input length $k = 5$. Let $\Sigma_{\text{orig}}$ and $\Sigma_{\text{alt}}$ be disjoint token sets with a bijection $\pi : \Sigma_{\text{orig}} \to \Sigma_{\text{alt}}$ and $\sigma_{\text{alt}} = \pi \circ \sigma_{orig} \circ \pi^{-1}$. We vary the token set $(\Sigma)$ to construct ID, OOD, and Mixed datasets:

$$\mathcal{D}_{\text{ID}}^{\text{tok}} = \{(x, S, \hat{x}) : x \in \Sigma_{\text{orig}}^5, \ S = (T_{\sigma_{\text{orig}}})\}, \quad \mathcal{D}_{\text{OOD}}^{\text{tok}} = \{(x, S, \hat{x}) : x \in \Sigma_{\text{alt}}^5, \ S = (T_{\sigma_{\text{alt}}})\},$$

$$\mathcal{D}_{\text{Mixed}}^{\text{tok}} = \{(x, S, \hat{x}) : x \in (\Sigma_{\text{orig}} \cup \Sigma_{\text{alt}})^5, \ S = (T_{\sigma_{\text{orig}} \sqcup \sigma_{\text{alt}}})\}.$$

- **Compositional Reasoning.** Fix the token set $\Sigma$ and graph mapping $\sigma$, input length $k = 5$, so any input $x \in \Sigma^5$. We compose operators $(S)$ to construct ID and OOD datasets:

$$\mathcal{D}_{\text{ID}}^{\text{comp}} = \{(x, S, \hat{x}) : \ S \in \{(T_\sigma^2), (R^2)\}\}, \quad \mathcal{D}_{\text{OOD}}^{\text{comp}} = \{(x, S, \hat{x}) : \ S \in \{(R, T_\sigma), (T_\sigma, R)\}\}.$$

### 4.3 MODEL CONFIGURATION

We experiment with small Llama-decoder-based transformers to enable efficient training and evaluation while retaining the architectural characteristics of modern LLMs. Specifically, our model uses a hidden dimension $d_{\text{model}} = 512$, intermediate feed-forward dimension $d_{\text{ff}} = 1376$, number of layers $L = 4$, and number of attention heads $h = 8$. We adopt rotary position embeddings with parameter $\theta_{\text{rope}} = 10,000$ and apply RMSNorm with $\epsilon = 10^{-6}$. The total number of parameters of the model is around 45 million.

## 5 EXPERIMENTS

### 5.1 REASONING DEPTH GENERALIZATION

A critical dimension of reasoning is the ability to dynamically adapt deductive depth to a problem's intrinsic complexity. For challenging tasks, a model must extrapolate beyond its training, composing longer and more intricate chains of thought. Conversely, a model trained only on complex problems can become brittle on simpler tasks where a briefer, more direct solution is optimal. Our reasoning depth generalization test investigates this flexibility: *Does GRPO generalize to both greater and shallower reasoning depths?*

**Experimental setup.** For deeper generalization, models are trained on 1- and 2-step tasks (ID) and evaluated on 3-step tasks (OOD). A symmetric experiment for shallower generalization defines 2- and 3-step tasks as ID and 1-step tasks as OOD. To analyze data distribution's impact, we pretrain models on mixtures ranging from purely ID data (e.g., 50% 1-step, 50% 2-step, 0% 3-step) to a uniform mix across all depths. Models are then supervise-finetuned on the ID dataset, and GRPO is applied using either ID or OOD data.

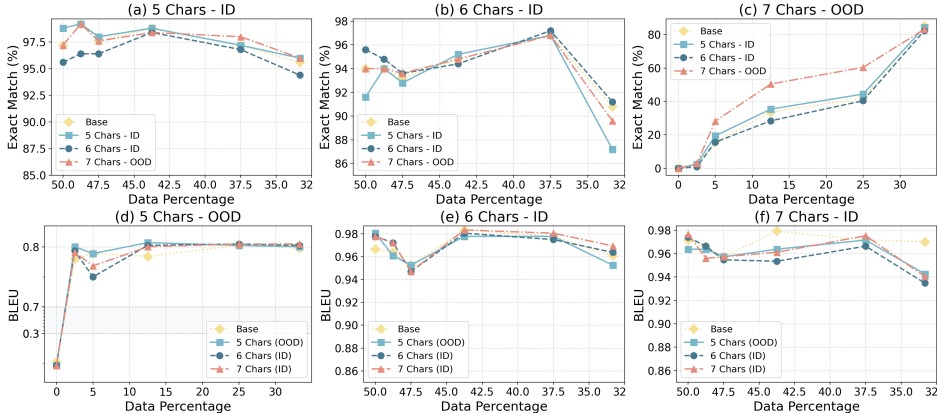

Figure 4: Input length generalization. Top row (a–c): models pretrained on shorter sequences (5–6 chars, ID) mixed with longer OOD sequences (7 chars, OOD). Bottom row (d–f): models pretrained on longer sequences (6–7 chars, ID) mixed with shorter OOD sequences (5 chars, OOD).

**Findings.** To ensure GRPO's effectiveness, a model must be pretrained on sequences of similar depth. For instance, in the deeper generalization setting (ID: 1- & 2-step, OOD: 3-step), when trained only on 1- and 2-step tasks, the model retains high ID accuracy but completely fails on 3-step tasks even after GRPO (Figure 3c). However, introducing 3-step data in pretraining boosts 3-step performance substantially, with exact match increasing by 8% to 39% as the OOD ratio grows from 2.5% to 33.3%. A smaller ID–OOD gap facilitates ID-OOD transfer: GRPO with 2-step data improves 3-step accuracy, whereas 1-step data does not produce gains (Figure 3c). In the shallower generalization setting (ID: 2- & 3-step, OOD: 1-step), the model again achieves 0% on OOD without exposure (Figure 3d). However, GRPO using 1-step OOD data markedly improves 1-step performance. This is possible because 1-step reasoning is an encoded sub-component of the more complex 2- and 3-step tasks seen during pretraining. The exact match improves by 6% to 36% as the OOD ratio grows from 0% to 33.3%. In both settings, th already-saturated ID depths show little to no improvement (Figure 3a,b,e,f). Collectively, these results show that GRPO does not extrapolate to unseen depths; it transfers when the OOD length has already been seen, and the magnitude of improvement scales with the proportion of similar data available during pretraining.

## 5.2 INPUT LENGTH GENERALIZATION

The capacity to handle inputs of varied and unforeseen lengths is a fundamental test of a language model's generalizability. Models must not only extrapolate to capture dependencies in documents longer than any seen during training but also remain effective on concise, information-dense queries. Our input length generalization test is designed to answer: *Does GRPO enable generalization to both longer and shorter inputs than those seen during training?*

**Experimental setup.** For generalization to longer inputs, 5- and 6-character sequences are ID, and 7-character sequences are OOD. The process involves pretraining on data mixes ranging from purely ID (50% 5-char, 50% 6-char, 0% 7-char) to a uniform ratio. We then fine-tune on ID data and apply GRPO with either ID-only or OOD-only data. A complementary experiment on generalization to shorter inputs defines 6- and 7-character sequences as ID and 5-character sequences as OOD.

**Findings.** For longer input generalization (ID: 5- & 6-char, OOD: 7-char), GRPO is ineffective if the model has zero prior exposure, failing to improve upon a 0% exact match. However, once the base model gains a foundation through pretraining with 7-char data, GRPO provides a significant gain, peaking with a 21% improvement when the pretraining mix contains 12.5% OOD data. Conversely, for shorter input generalization (ID: 6- & 7-char, OOD: 5-char), increasing the OOD pretraining data improves the BLEU scores of the base model but fails to improve the exact match. This is due to a specific failure mode where the model does not generate the correct number of padding tokens (see Appendix D.1), and GRPO cannot recover an exact match in this case. And since the BLEU scores are already high for both ID and OOD with a small 2.5% 7-char data exposure, GRPO finetuning offers no improvement on BLEU scores (Figure 4d,e,f). This demonstrates GRPO's core limitation: it is not does not work for completely unaligned models, nor is it useful for already optimized tasks.

Instead, it excels in a moderate gap, when the base model is partially aligned but still has a clear and bridgeable gap to optimal performance.

## 5.3 Token Representation Generalization

An effective reasoning model should grasp the underlying logic of a task, regardless of its surface-level presentation. For example, the core logic for implementing an algorithm is the same whether one is writing code in Python or Java. A model that can perform well in these scenarios demonstrates an ability to focus on this core logic rather than memorizing superficial patterns in the input. This motivates our input representation generalization test, which examines the following question: *Does GRPO enable generalization to different representations that share the same underlying structures?*

**Experimental setup.** We test generalization to new token representations while preserving the underlying task logic. The ID task involves graph traversal using a graph built from the original token set, while the OOD task applies identical logic to an isomorphic graph built from an alternative token set. Models are pretrained on data mixtures ranging from 100% ID data to a 50/50 ID-OOD split. After fine-tuning on ID-only data, we apply GRPO using either ID or OOD data to measure generalization. Finally, we evaluate robustness by mixing ID inputs with 1-3 OOD tokens.

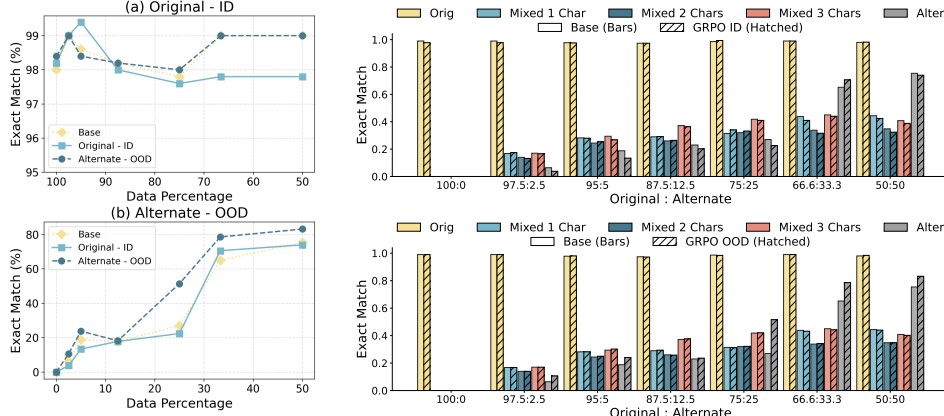

Figure 5: Token representation generalization.

Figure 6: Performance on mixed inputs.

**Findings.** The base model pretrained only on ID tokens achieves 0% exact match on OOD tokens, even after GRPO (Figure 5b). Only incorporating OOD tokens in pretraining steadily improves OOD base performance, and GRPO then becomes effective, boosting performance by up to 30% with gains peaking at 25% OOD data. By contrast, GRPO on ID data cannot raise ID performance, as it is already 98%, and even degrades OOD performance (Figure 5a). Moreover, GRPO's benefit is not robust against token mixing: when ID inputs are mixed with even a single OOD token, base and GRPO models perform equally poorly (Figure 6). This shows GRPO adapts to specific token vocabularies rather than instilling robustness against token mixing. Therefore, GRPO heavily relies on sharpening biases from pretraining tokens: a larger OOD token ratio during pretraining induces a larger improvement until near saturation, but GRPO's improvement does not translate to token mixing, even when the inputs share an identical task structure.

## 5.4 Compositional Reasoning Generalization

Sophisticated problem-solving demands the ability to combine learned skills in novel sequences. For instance, an LLM agent trained on atomic API calls may need to chain them, dynamically using the output of one function as the input for another to solve a more complex task. We therefore investigate: *Does GRPO enable generalization to novel compositions of learned operations?*

**Experimental setup.** We test compositional generalization using *traversal* and *shift* operations, with all tasks fixed to a two-step process to control for length. ID tasks are single operations (*traversal*-only or *shift*-only), while the OOD task is their composition: *shift*+*traversal*. Models are pretrained on data mixtures ranging from purely ID data (50% *traversal*-only, 50% *shift*-only) to a uniform mix. After fine-tuning on the ID set, we apply GRPO using either ID-only or OOD-only data. A similar experiment uses the reverse composition, *traversal* + *shift*, as the OOD task.

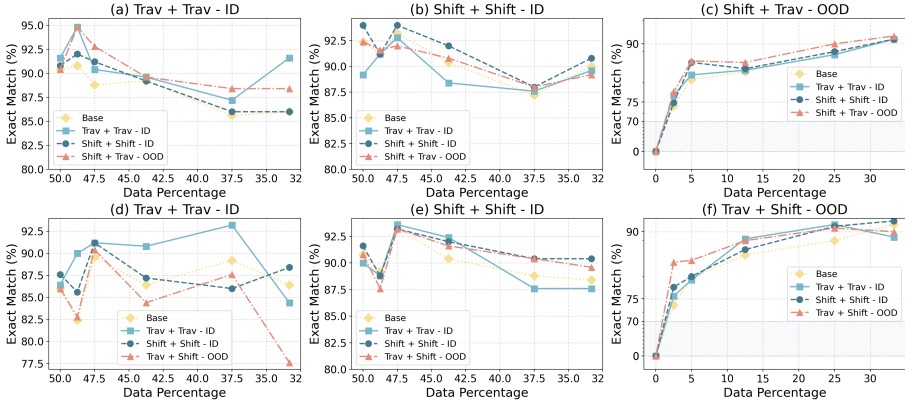

Figure 7: Compositional reasoning generalization with varying pretraining ratios. Top row (a-c): models pretrained on separate tasks (ID: trav+trav and shift+shift) are progressively mixed with the composite task (OOD: shift+trav). Bottom row (d-f): OOD task is trav+shift instead.

**Findings.** A base model pretrained only on singular ID tasks (*traversal*-only, *shift*-only) fails completely on the composite OOD task with 0% exact match, and GRPO provides no improvement. Composite performance improves only when composite data are included in pretraining; even 2.5% OOD exposure yields 75% exact match. With this foundation, GRPO becomes useful, showing up to +6% for *shift+traversal* at 5% OOD (Figure 7c) and +10% for *traversal+shift* at 2.5% OOD (Figure 7f). Thus, modest OOD ratios create sufficient task alignment for GRPO to enhance composite reasoning. Conversely, GRPO on ID data mainly reinforces ID performance (Figure 7a,b,d,e), with marginal benefits to OOD due to shared skills (Figure 7c,f). Overall, GRPO cannot induce composite reasoning from singular-task pretraining alone, but once the model sees even limited composite data, GRPO can amplify this capability by training on either singular or composite tasks.

# 6 SUPPLEMENTARY ANALYSIS ON GENERALIZATION

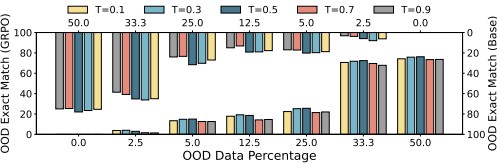

Figure 8: Exact match with varying temperatures in token generalization.

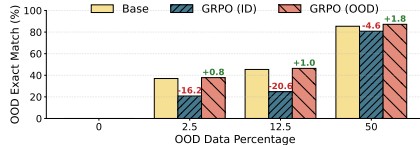

Figure 9: Exact match with 0.5B model in token generalization.

**The effects of temperature on OOD.** For token representation generalization, as we do not see any ID-OOD transfer, we further examine the influence of sampling temperature in this setting. We vary the sampling temperature from 0.1 to 0.9 for evaluation and observe that varying the temperature leads to small variance but still does not promote ID-OOD transfer, showing that our findings are consistent across temperatures.

**The effect of different parameter size.** We scale up the base model to 0.5B parameters to test token representation generalization. Consistent with previous findings, GRPO provides no gains without exposure to alternative tokens. Only by introducing them during pre-training does GRPO improve performance. However, we still observe no ID-OOD transfer, but rather significant forgetting. This indicates that token generalization remains a difficult challenge, even for larger models.

# 7 CONCLUSIONS

This work reframes GRPO as a conservative reweighting mechanism that sharpens existing inductive biases rather than discovering new reasoning. We prove it operates based on the base model's distribution, making it unable to generate solutions outside the scope. We conduct controlled experiments with transformers trained from scratch to confirm this boundary: OOD generalization occurs only when the target task aligns with the model's inherent biases. A smaller alignment gap yields larger gains until saturation, but without pretraining on the OOD task, GRPO brings no improvement. These findings explain GRPO's inconsistent gains—its success depends on task–bias overlap—highlighting the need for algorithms that explicitly expand the solution space.

REPRODUCIBILITY STATEMENT

We have taken several measures to ensure the reproducibility of our results. All hyperparameters for training and evaluation are documented in Appendix B. The main paper provides a clear description of our theoretical framework, with complete proofs presented in Appendix C. Our experimental setup is described in detail in Section 4, where we define the synthetic datasets, task formulations, and ID–OOD splits, with representative examples shown in Table 1.

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

CONTENTS OF THE APPENDIX

## A CORRELATION BETWEEN GRPO IMPROVEMENT AND PERPLEXITY

We show that for Qwen-2.5-7B, its gains from GRPO on various datasets are inversely correlated with perplexity scores: a smaller perplexity score corresponds to a larger gain (Table 2).

## B HYPERPARAMETERS

### B.1 GRPO BENCHMARKING

The hyperparameters used for benchmarking GRPO are consistent across all datasets and models: learning rate $= 3\text{e}-6$, batch size $= 64$, samples per prompt $= 8$, warmup ratio $= 0.1$, KL coefficient $= 0.005$, and training steps $= 30$. Evaluations are conducted with temperature $t = 0.3$, nucleus sampling $p = 0.8$, and maximum sequence length $= 3072$ tokens.

| Task | PPL | Before | After | $\Delta$ Acc. |
|------|-----|--------|-------|------|
| Math | 2.56 | 45.20 | 76.40 | 31.20 |
| CommonsenseQA | 16.19 | 58.20 | 81.90 | 23.70 |
| MedMCQA | 17.63 | 46.78 | 57.25 | 10.47 |
| PharmDB | 41.35 | 57.93 | 63.10 | 5.17 |

Table 2: Qwen2.5 7B: perplexity and accuracy before/after GRPO. $\Delta$ is the improvement in percentage points.

### B.2 GENERALIZATION TESTS

For the generalization tests, the training hyperparameters are detailed in Table 3. Corresponding evaluations are performed using a temperature of $t = 0.1$, nucleus sampling with $p = 0.8$, and a maximum sequence length of 256 tokens.

| Test | Stage | LR | Batch | Samples | Warmup | KL | Epoch / Steps |
|------|-------|-----|-------|---------|--------|-----|---------------|
| | Pretraining | 1e−3 | 131072 | – | 0.1 | 0.005 | 1 Epoch |
| Reasoning Depth | SFT | 2e−4 | 64 | – | 0.1 | 0.005 | 1 Epoch |
| | GRPO | 3e−6 | 64 | 8 | 0.1 | 0.005 | 60 Steps |
| | Pretraining | 1e−3 | 131072 | – | 0.1 | 0.005 | 1 Epoch |
| Input Length | SFT | 2e−4 | 64 | – | 0.1 | 0.005 | 1 Epoch |
| | GRPO | 3e−6 | 64 | 8 | 0.1 | 0.005 | 60 Steps |
| | Pretraining | 1e−3 | 131072 | – | 0.1 | 0.05 | 1 Epoch |
| Token Representation | SFT | 1e−4 | 64 | – | 0.1 | 0.05 | 1 Epoch |
| | GRPO | 5e−6 | 64 | 8 | 0.1 | 0.05 | 60 Steps |
| | Pretraining | 1e−3 | 131072 | – | 0.1 | 0.005 | 1 Epoch |
| Compositional Reasoning | SFT | 2e−4 | 64 | – | 0.1 | 0.005 | 1 Epoch |
| | GRPO | 3e−6 | 64 | 8 | 0.1 | 0.005 | 60 Steps |

Table 3: Hyperparameters for each generalization test.

## C PROOFS

Throughout, we assume a discrete completion space $\mathcal{Y}$ and write $C_x = \{y \in \mathcal{Y} : R(x, y) = 1\}$ and $Q(x) = q(C_x \mid x)$. Expectations over $x$ are taken with respect to the prompt distribution $d$ and are omitted when clear. We work pointwise in $x$ and suppress the dependence on $x$ whenever it does not cause confusion. Let $a := e^{\beta^{-1}} > 1$ and define

$$f_\beta(Q) := \frac{aQ}{(1 - Q) + aQ} \quad \text{and} \quad g_\beta(Q) := f_\beta(Q) - Q$$

for $Q \in [0, 1]$.

## C.1 PROOF OF THEOREM 1 (CLOSED FORM, MONOTONICITY, SUPPORT PRESERVATION, SATURATION)

**Closed form.** Fix any $x$. Consider

$$\max_{\pi(\cdot|x)} \left\{ \sum_{y \in \mathcal{Y}} \pi(y \mid x) R(x, y) - \beta \sum_{y \in \mathcal{Y}} \pi(y \mid x) \log \frac{\pi(y \mid x)}{q(y \mid x)} \right\} \quad \text{s.t.} \quad \sum_{y} \pi(y \mid x) = 1, \ \pi(\cdot \mid x) \geq 0.$$

The Lagrangian (with multiplier $\lambda$ for the simplex constraint) is

$$\mathcal{L}(\pi, \lambda) = \sum_{y} \pi(y) R(x, y) - \beta \sum_{y} \pi(y) \log \frac{\pi(y)}{q(y)} + \lambda \Big( \sum_{y} \pi(y) - 1 \Big).$$

Stationarity gives, for every $y \in \mathcal{Y}$,

$$0 = \frac{\partial \mathcal{L}}{\partial \pi(y)} = R(x, y) - \beta \Big( \log \frac{\pi(y)}{q(y)} + 1 \Big) + \lambda \quad \Longleftrightarrow \quad \log \frac{\pi(y)}{q(y)} = \beta^{-1} R(x, y) + c,$$

where $c = (\lambda - \beta)/\beta$ is independent of $y$. Exponentiating and normalizing over $y$ yields the unique optimum

$$\pi_\beta^\star(y \mid x) = \frac{q(y \mid x) \exp\big(\beta^{-1} R(x, y)\big)}{\sum_{y'} q(y' \mid x) \exp\big(\beta^{-1} R(x, y')\big)}.$$

With binary rewards, writing $a = e^{\beta^{-1}}$, we obtain

$$\pi_\beta^\star(C_x \mid x) = \frac{\sum_{y \in C_x} q(y \mid x) a}{\sum_{y \notin C_x} q(y \mid x) + \sum_{y \in C_x} q(y \mid x) a} = \frac{a \, Q(x)}{(1 - Q(x)) + a \, Q(x)} = f_\beta\big(Q(x)\big).$$

**Monotonicity, support, saturation.** For $f_\beta(Q) = \dfrac{aQ}{1 + (a - 1)Q}$ we have

$$f_\beta'(Q) = \frac{a}{\big(1 + (a - 1)Q\big)^2} > 0 \quad \text{for all } Q \in [0, 1],$$

so $f_\beta$ is strictly increasing. If $Q = 0$ then $f_\beta(0) = 0$, showing *support preservation*: whenever $q(C_x \mid x) = 0$, we also have $\pi_\beta^\star(C_x \mid x) = 0$. Finally, $\lim_{Q \uparrow 1} f_\beta(Q) = 1$, i.e., improvement saturates as $Q \to 1$.

## C.2 PROOF OF COROLLARY 1 (MARGINAL-GAIN MONOTONICITY)

Define the marginal-gain functional

$$\mathsf{MGain}_\beta(q) := \mathbb{E}_{x \sim d}\big[g_\beta(Q(x))\big], \qquad g_\beta(Q) = f_\beta(Q) - Q, \quad f_\beta(Q) = \frac{aQ}{1 + (a - 1)Q}, \quad a = e^{\beta^{-1}} > 1.$$

A direct calculation gives

$$g_\beta'(Q) = f_\beta'(Q) - 1 = \frac{a}{\big(1 + (a - 1)Q\big)^2} - 1.$$

Thus $g_\beta'(Q) \geq 0$ exactly on $Q \in [0, \tau(a)]$ where

$$\tau(a) := \frac{\sqrt{a} - 1}{a - 1} \quad \text{since} \quad g_\beta'(Q) \geq 0 \iff 1 + (a - 1)Q \leq \sqrt{a}.$$

Consequently, $g_\beta$ is pointwise increasing on $[0, \tau(a)]$ and decreasing thereafter. If two base policies $q, q'$ satisfy $Q'(x) \geq Q(x)$ for all $x$ and $Q(x) \leq \tau(a)$ for all $x$ (low-overlap regime), then by pointwise monotonicity on this interval,

$$g_\beta\big(Q'(x)\big) \geq g_\beta\big(Q(x)\big) \quad \text{for all } x,$$

and taking expectations yields $\mathsf{MGain}_\beta(q') \geq \mathsf{MGain}_\beta(q)$. This shows that below some point, a smaller distribution gap produces a larger marginal gain.

Finally, the saturation of marginal gains at high overlap follows directly within the same argument: since $f_\beta(Q) \to 1$ as $Q \to 1$, we have

$$g_\beta(Q) = f_\beta(Q) - Q \longrightarrow 1 - 1 = 0,$$

and, for all $Q \in [0, 1]$,

$$g_\beta(Q) = \frac{aQ}{1 + (a-1)Q} - Q = \frac{Q(1-Q)(a-1)}{1 + (a-1)Q} \le 1 - Q,$$

so $g_\beta(Q) \le 1 - Q \to 0$ as $Q \to 1$. This shows the "diminishing due to saturation" behavior.

### C.3  PROOF OF PROPOSITION 1 (SMALL-MASS LINEAR BOUND)

From the closed form,

$$\pi_\beta^\star(C_x \mid x) = \frac{a\,Q(x)}{(1 - Q(x)) + a\,Q(x)} \le a\,Q(x),$$

since the denominator is at least 1. Therefore

$$\pi_\beta^\star(C_x \mid x) - Q(x) \le a\,Q(x) - Q(x) = (a-1)\,Q(x) = \left(e^{\beta^{-1}} - 1\right)Q(x),$$

which is linear in $Q(x)$ and tight to first order as $Q(x) \to 0$.

### C.4  PROOF OF PROPOSITION 2 (TOKEN FLOORS AND EXPONENTIALLY SMALL SEQUENCE MASS)

We formalize the statement in a worst-case sense.

**Setup.**  Let the vocabulary size be $V < \infty$ and suppose that at every decoding step $t = 1, \ldots, T$ and for every history, each token $a$ has probability bounded below by a fixed floor $\eta \in (0, 1/V]$:

$$q_{\mathrm{tok}}(a \mid \mathrm{history}_t) \ge \eta \quad \text{for all tokens } a \text{ and histories.}$$

**Existential worst case.**  Under only this floor constraint, there exist base policies $q$ for which, for any fixed set of correct completions $C_x$ of length $T$,

$$Q(x) = q(C_x \mid x) \le |C_x|\,\eta^T.$$

Construct $q$ stepwise as follows. At each position $t$ and for each history that occurs along any $y \in C_x$, assign probability exactly $\eta$ to the next token prescribed by that correct path, and distribute the remaining mass (so that all tokens still receive at least $\eta$) across the other $V$ tokens. This ensures that every $y \in C_x$ has

$$q(y \mid x) = \prod_{t=1}^{T} q_{\mathrm{tok}}(y_t \mid \mathrm{history}_t) = \eta^T,$$

and hence $Q(x) = \sum_{y \in C_x} q(y \mid x) = |C_x|\,\eta^T$.

**Consequence for inverse temperature.**  Combining Proposition 1 with the above bound,

$$g_\beta(x) = f_\beta(Q(x)) - Q(x) \le (a-1)\,Q(x) \le (a-1)\,|C_x|\,\eta^T.$$

To achieve a target increment $\mathrm{MGain}_\beta(x) \ge \varepsilon$ (for fixed $|C_x|$ and $\eta$), it is necessary that

$$a - 1 \ge \frac{\varepsilon}{|C_x|\,\eta^T} \quad \Longleftrightarrow \quad \beta^{-1} = \log a \ge \log\left(1 + \frac{\varepsilon}{|C_x|\,\eta^T}\right).$$

As $T$ grows with $\varepsilon, |C_x|, \eta$ fixed, the right-hand side is $\Omega\big(T \log(1/\eta)\big)$, yielding the claimed scaling.

## D  CASE STUDIES

We present qualitative analysis for each type of generalization test.

| Prompt | Generated | Expected |
|--------|-----------|----------|
| 6CI4R+++ <trav> | => JG2RV++ | => JG2RV+++ |
| D29UO+++ <trav> | => IHS1K++ | => IHS1K+++ |
| NEIAE+++ <trav> | => OO2FO++ | => OO2FO+++ |
| R751K+++ <trav> | => VQADU++ | => VQADU+++ |
| 2S521+++ <trav> | => HEAHD++ | => HEAHD+++ |

Table 4: Examples showing failure to generalize to shorter input lengths even when using a uniform ratio (33% 5-char, 33% 6-char 33% 7-char sequences) and applying GRPO with 7-character OOD data.

| Prompt | Base | GRPO |
|--------|------|------|
| kulvq$\gamma$ <trav> | => oy$\gamma$h$\alpha$k <trav> => oh$\lambda$hokcjckqvqj | => oy$\gamma$h$\alpha$k |
| $\delta$cpa$\alpha$g <trav> | => wv$\lambda$tall | => wv$\lambda$tal |
| inqckj <trav> | => sx$\alpha$vohc | => sx$\alpha$voc |
| fbu$\alpha\beta$h <trav> | => jqyagrh6 <trav> => rtlrnk | => jqyagr |
| $\lambda$jt$\lambda$i$\gamma$ <trav> | => mc$\theta$mskc | => mc$\theta$msk |

Table 5: Examples showing that GPPO corrects the base model in token representation generalization.

## D.1 INPUT LENGTH GENERALIZATION

For shorter length generalization test (ID is 6- & 7-character and OOD is 5-character), we observe that the model continuously fails to generate the correct number of padding tokens at the end, even when using a uniform ID-OOD ratio during pretraining and applying GRPO with 1-step OOD data (Table 4). This shows that the model does not inherently recognize shorter inputs, and it is also difficult to correct this bias with GRPO. This hints that SFT on OOD data is directly needed.

## D.2 TOKEN REPRESENTATION GENERALIZATION

We observe that the base model struggles when applying two distinct learned functions (shift and traversal) to a single prompt. The base model frequently fails by executing only one of the two functions, often incompletely, or by making critical errors when attempting both. In contrast, when including OOD data within pretraining and applying GRPO with composite data, GRPO consistently succeeds at these compositional tasks, correctly applying both the shift and traversal operations to generate the full correct chain.

## D.3 COMPOSITIONAL REASONING GENERALIZATION

We observe that the base model struggles when applying two distinct learned operations (shift and traversal) to a single prompt, even when the composite data are included in pretraining. The base model frequently fails by executing only one of the two functions, often incompletely, or by making critical errors when attempting both. When applying GRPO with composite data, GRPO reduces these errors, correctly applying both the shift and traversal operations to generate the full correct chain.

## E THE USE OF LLMS

We employed Google's Gemini 2.5 Pro and OpenAI's GPT-5 as writing assistance tools during the preparation of this manuscript. Their role was exclusively for language refinement, such as

| Prompt | Base | GRPO |
|---|---|---|
| ceksδg `<trav>` | => vpoewlwδcjλcvg `<trav>`
=> vcvhvlλ | => vpewlwlγλwλlgjγ `<trav>`
=> γlcγkwλγlγkγλ `<trav>`
=> rγk6kwλ6jλkγ `<trav>`
=> J6pcwk |
| tiwwhs `<trav>` | => θsεre6 | => θsεreλmλεrje `<trav>`
=> stupn6njcαδrcjθj `<trav>`
=> rnv6c |
| γλθwad `<trav>` | => kmdεtiδwλmλgvkh `<trav>`
=> λmlo `<trav>`
=> mxfδh6hkdltγθj | => kmdεtiδwλmλgvkh `<trav>`
=> λmlohv6khδwδwγ `<trav>`
=> m6wkεmλ `<trav>`
=> λblfm |

Table 6: Examples showing that GRPO fails to correct the base model in token representation generalization and in turn increases the length of incorrect responses.

| Prompt | Base | GRPO |
|---|---|---|
| JEL3E `<shift>` `<trav>` | => EL3E | => EL3EJ `<trav>`
=> O3TOC |
| ANANE `<shift>` `<trav>` | => NANE | => NANEA `<trav>`
=> 0F0OF |
| HTIUV `<shift>` `<trav>` | => TIUV4 `<trav>`
=> 4218R | => TIUVH `<trav>`
=> 4218L |
| A55H2 `<shift>` `<trav>` | => 55H2A `<trav>`
=> AAHLF | => 55H2A `<trav>`
=> AALHF |
| 1BFFN `<shift>` `<trav>` | => BFFN1 `<trav>`
=> MPP0 | => BFFN1 `<trav>`
=> MPP0D |

Table 7: Examples showing that GRPO can correct the base model in compositional reasoning generalization.

improving readability and rephrasing for clarity in an academic writing style. This usage aligns with standard academic practices for language polishing.

