# OpenReview forum: "Can GRPO Help LLMs Transcend Their Pretraining Origin?"
_ICLR.cc/2026/Conference — Submitted to ICLR 2026_

### Official Review · Reviewer_3aW7 · 2025-10-28

**Soundness:** 3
**Presentation:** 3
**Contribution:** 2
**Rating:** 4
**Confidence:** 4

**Summary:**

This paper looks at whether the GRPO algorithm, used in reinforcement learning to improve LLMs, actually helps models think better or just makes them more confident in what they already know. The authors show, both theoretically and through controlled experiments, that GRPO doesn’t teach models new reasoning—it only boosts patterns already present from pretraining. They train small models from scratch on synthetic datasets to test how GRPO handles different types of new situations, like longer reasoning chains or unfamiliar tokens, and find that it only works when the new task is similar to what the model has already seen.

**Strengths:**

* By training small transformers from scratch using synthetic datasets, the authors effectively avoid the complexities and opacity of large, pre-trained corpora.
* The results have immediate relevance for the design of reasoning-focused training schemes.

**Weaknesses:**

* The theoretical analysis assumes the presence of a KL-divergence regularization term in the RLVR optimization objective, which bounds the learned policy close to the base model’s distribution. While this assumption is standard in GRPO formulations, it may overconstrain the analysis. In practical RLVR setups with verifiable rewards, the risk of reward hacking is absent, and recent work shows that removing the KL penalty can yield better performance by allowing broader exploration [1]. Consequently, the “conservative reweighting” interpretation might be an artifact of the imposed KL constraint rather than a fundamental property of all RLVR variants, meaning the paper’s theoretical conclusions may not generalize to algorithms that optimize under looser or KL-free objectives.
* While the theory explains what GRPO cannot do, there is little introspection into how model representations change or fail to change during RLVR. Visualization or probing of hidden states could strengthen the argument.
* The figures throughout the paper do not include standard deviation or confidence intervals, and many plotted points appear extremely close to each other. This makes it difficult to judge the statistical reliability of the reported trends and raises concerns that some observed patterns could be artifacts of noise rather than meaningful effects.

[1] Liu et al, Understanding R1-Zero-Like Training: A Critical Perspective

**Questions:**

* See weaknesses

---

### Official Review · Reviewer_gudw · 2025-10-29

**Soundness:** 2
**Presentation:** 3
**Contribution:** 1
**Rating:** 2
**Confidence:** 4

**Summary:**

The paper studies whether GRPO (RL with verifiable rewards) can push reasoning LLMs beyond what is present in their pretrained distribution. The authors (i) prove that the GRPO optimum is an exponential reweighting of the base policy (hence "conservative"), with gains controlled by the base model’s correct-mass (Q(x)); and (ii) validate this in controlled synthetic settings (reasoning depth, input length, token remapping, composition) by training small transformers from scratch. Empirically, OOD improvements appear only when OOD overlaps with pretraining, and they saturate as base performance saturates. The paper concludes that GRPO “sharpens biases” rather than discovering new reasoning outside pretraining support.

**Strengths:**

1. **Crisp theoretical framing.** Formalizes GRPO as conservative exponential tilting and cleanly relates attainable gains to the base model’s correct-mass (Q(x)).
2. **Controlled synthetic testbed.** From-scratch training and clean ID/OOD splits reduce confounding and make the failure modes reproducible.
3. **Consistent theory and empirics.** Across four OOD axes, results align with the support-limited view: no OOD gains without overlap; saturation with high base alignment.
4. **Useful negative result.** Helps explain why RLVR sometimes fails on hard OOD when the base cannot produce reward-earning samples.

**Weaknesses:**

1. **Limited Scope and Misattribution of Failure.** The paper frames its central negative result ("cannot transcend pretraining support") as a specific limitation of GRPO. However, the experimental setup, characterized by near-zero support for correct trajectories (Q(x) ≈ 0), sparse binary rewards, and a short training horizon, constitutes a worst-case scenario for any on-policy policy-gradient (PG) algorithm lacking a dedicated exploration mechanism. The observed failure is more indicative of the fundamental challenge of learning from zero reward signal, rather than an inherent flaw unique to GRPO. It is highly probable that other PG methods, such as PPO, would exhibit similar stalling behavior under these conditions. The claim should therefore be re-scoped to "on-policy RLVR in low-support regimes" instead of being narrowly attributed to GRPO.
2. **Overlooks Contradictory Evidence from Recent Literature.** The paper's strong conclusion that RLVR is fundamentally incapable of discovering novel solutions is challenged by recent findings. For instance, ProRL [1] demonstrates that prolonged RL training (thousands of steps), coupled with techniques to maintain exploration (e.g., reference policy resets), and diverse tasks set can indeed expand reasoning boundaries, enabling models to solve problems where the base model had pass@k ≈ 0. Similarly, [2] shows that RL can successfully teach models new compositional skills not present in the base model. The conclusions of the current paper may thus be an artifact of its short-training setup, rather than a fundamental limitation of RLVR itself.
3. **Conflation of Algorithmic and Architectural Limitations.** The paper attributes all observed generalization failures to the GRPO algorithm. However, certain failures, particularly in length generalization, are well-known architectural limitations of Transformer models (e.g., related to positional encodings). The authors do not disentangle these confounding factors, making it difficult to isolate the true effect of the RL objective. The analysis would be stronger if it controlled for or at least discussed the role of model architecture in these specific failure modes. The work of [3] also offers a more nuanced, trajectory-level view where RL "squeezes" distributions, which is a more precise description than the blanket "impossibility" claim made here.

[1] Liu et al., 2025. ProRL: Prolonged Reinforcement Learning Expands Reasoning Boundaries in LLMs. arXiv:2505.24864.

[2] Yuan et al., 2025. From (f(x)) and (g(x)) to (f(g(x))): LLMs Learn New Skills in RL by Composing Old Ones. arXiv:2509.25123.

[3] Matsutani et al., 2025. RL Squeezes, SFT Expands: A Comparative Study of Reasoning LLMs. arXiv:2509.21128.

**Questions:**

1. **Specificity to GRPO:** Have you considered testing other RLVR algorithms like PPO in your synthetic testbed? If the results are similar, wouldn't it be more accurate to frame the paper's conclusions around the challenges of on-policy RL in sparse-reward, zero-support settings, rather than singling out GRPO?
2. **Reward Shaping and Curriculum:** The problem of zero initial support is often addressed using techniques like reward shaping or curriculum learning [4]. How does your theoretical framework account for scenarios where a curriculum of progressively harder tasks provides an initial learning signal, potentially allowing the model to "bootstrap" its way to solving previously unsolvable problems?
3. **Theoretical Scope:** Does your theoretical analysis (specifically Theorem 1 and its consequences) hold for more complex reward schemes, such as token-level, or process-based rewards, which are also common in RLVR?

[4] Parashar, S., Gui, S., Li, X., Ling, H., Vemuri, S., Olson, B., ... & Ji, S. (2025). Curriculum Reinforcement Learning from Easy to Hard Tasks Improves LLM Reasoning. arXiv preprint arXiv:2506.06632.

---

### Official Review · Reviewer_6m7p · 2025-11-01

**Soundness:** 2
**Presentation:** 3
**Contribution:** 2
**Rating:** 2
**Confidence:** 3

**Summary:**

This paper studies whether Group Relative Policy Optimization (GRPO) and Reinforcement Learning with Verifiable Rewards (RLVR) in general amplifies already existing patterns rather than introduce novel problem-solving skills. The authors develop theoretical apparatus to prove that GRPO reweighs the base model probabilities instead of changing the behaviour completely; to validate their theoretical conclusions, authors conduct a controlled experiment to test various GRPO generalization properties, showing that RLVR-trained model is unable to generate solutions that are not already contained in the base model.

**Strengths:**

S1. Motivation presented clearly and efficiently with nicely collected related literature.

S2. An idea to isolate specific properties of GRPO at small scale is clever, and the experiments proposed in the paper directly tackle the question at hand and are executed well.

S3. Theoretical analysis and conceptual framework (notions of low mass, alignment principle etc.) are interesting and seems to be useful.

**Weaknesses:**

W1. It is unclear how the experimental results from the paper coincide with the real-world RLVR practice; the setup is very synthetic and is based on operations with fixed-length strings, which is not quite natural for multi-task models that combine various problem-solving skills and memorization during reasoning - in this setup model must implement some kind of algorithm, instead of learning general skills and acquiring cognitive capabilities. ID and OOD splits are somewhat artificial - modern RLVR setups try to maximize the diversity of tasks and their properties during training, without explicitly dividing them into some kind of easy and hard sets (unless for curriculum learning). The reason is understandable - it allows for performing the theoretical analysis of the GRPO algorithm, but the generalization of conclusions is questionable due to imposed constraints on the tasks, dataset construction procedures and tiny models.

W2. RL training seems to be of very small scale - only 60 steps. Recent results suggest that GRPO-trained models are able to learn novel reasoning skills at large training scale [1], which contradicts the premise of the work under review about the shallow reweighting; perhaps, this reweighting is still valid, but appears to be much more complicated at scale. The pretraining also seem to be of small scale in a sense that it does not allow for vast amount of patterns and knowledge to be memorized and applied during reasoning afterwards.

W3. Each experiment in section 5 is preceded by high-level explanation of some cognitive capabilities of some agent; however, the experiments are not directly related to these capabilities, producing a discrepancy that makes experiments seem less relevant for answering the main question. For example, in lines 383-385 a motivation is presented that the agent should be able to solve the problem regardless of it's surface-level representation and it should have the abstraction ability; the following experiment tests whether the model would work well if it is trained on graphs formed from one vocabulary and tested on isomorphic graphs written in another vocabulary, and the result is obtained that GRPO fails to achieve this goal. The reason could lie in the fact that the model is very small, task is very constrained and formal, and the setting suggest algorithm learning rather than general-purpose model learning with emergence and involvement of cognitive capabilities; i.e., the problem is not in GRPO or RLVR, but in the task design and neural network learning principles - the model just have never saw the tokens new graphs are build of, and therefore cannot predict anything about them (as if model has never saw a word in Chinese before, but we prompted it in Chinese). The same goes for other experiments, making the problem raised in W1 more pronounced.

W4. As a minor weakness, sometimes paper overly dense in its mathematical formulations (e.g. section 4), or in results reporting (all Findings paragraphs in section 5), making it harder to follow and understand the global picture.

**Questions:**

Q1. Can you speculate if the results of this paper will change when we change the structure of training batches in GRPO (e.g. by using curriculum learning and balancing the difficulty, so that e.g. half of the examples are correctly solved), or is it the setup GRPO would excel at (as in lines 378-379)?

Q2. How does your apparatus explains the spurious reward results, where even random reward could lead to successful improvement on benchmarks? It is primarily attributed to the structure of Qwen family pretraining dataset, but I wonder if you could provide more insights on this.

[1] ProRL: Prolonged Reinforcement Learning Expands Reasoning Boundaries in Large Language Models, Liu et al., 2025

---

### Official Review · Reviewer_B8bh · 2025-11-01

**Soundness:** 1
**Presentation:** 2
**Contribution:** 1
**Rating:** 2
**Confidence:** 3

**Summary:**

The paper provides a theoretical and empirical analysis of Group Relative Policy Optimization (GRPO) for large language models (LLMs). Key points:

- GRPO is shown to be a conservative reweighting scheme. It can only amplify patterns already present in the pretrained model. It cannot discover or generate truly novel solutions outside the model’s initial capabilities.
- Theoretical results prove GRPO’s effect is strictly limited by the pretrained model’s “correct mass” for any task. If the base model assigns zero or negligible probability to the correct solution, GRPO cannot improve performance on that task.
- Controlled experiments train transformer models from scratch on synthetic data to isolate and test four axes of generalization: reasoning depth, input length, token representation, and compositional reasoning.
- Main empirical finding: GRPO enables out-of-distribution generalization only when the target task aligns with the pretrained model’s biases or has partial overlap in the training distribution. Gains on in-distribution tasks diminish as performance saturates. GRPO does not induce new reasoning abilities, but sharpens existing ones.
- Paper concludes that GRPO is not a universal reasoning enhancer. Its utility is as a tool for refining pretrained biases, not for enabling models to transcend their original training. The results suggest the need for algorithms that can explicitly expand a model’s solution space beyond its pretraining origin.

**Strengths:**

- The paper’s originality lies in combining theory and controlled experiments (with models trained from scratch on synthetic data) to reveal precise limits on GRPO’s ability to support out-of-distribution generalization. Theoretical proofs clarify that GRPO is a conservative reweighting scheme, unable to discover solutions absent from the pretrained model’s distribution.
- The theoretical analysis is mathematically rigorous and fully proved. Experiments are cleanly designed. Using synthetic data to avoid confounding factors enables direct observation of when and why GRPO fails or succeeds. Results consistently validate the theory: GRPO’s gains require overlap between pretraining and target tasks, with no effect on completely novel or saturated tasks.
- The paper defines all terms and settings explicitly, visualizes experimental results, and ties empirical evidence directly to theoretical claims. Experimental axes (reasoning depth, input length, token representation, compositionality) are clearly described and systematically explored.
- The findings have a sufficient impact for the field. They show that GRPO (and similar RL post-training methods) cannot enable true generalization beyond what the model already “knows” from pretraining. The results explain inconsistent improvements seen across domains and motivate a shift towards developing algorithms that can expand rather than sharpen a model’s existing solution set.

**Weaknesses:**

1. **Theoretical Limitations**
    - The “exponential-tilting” derivation treats GRPO as closed-form reweighting, omitting effects of stochastic gradient updates, noise, or clipping found in practice.
    - Theoretical analysis omits practical factors: variance reduction, group normalization, and advantage estimation which affect GRPO optimization in real settings.
2. **Experimental Design Weaknesses**
    - All core experiments use synthetic toy data, not natural-language or semantically meaningful reasoning tasks. The setup removes the semantic and linguistic complexity critical for true OOD generalization in LLMs. No evaluation on real-world LLM reasoning datasets (GSM8K, MMLU), so claims about reasoning generalization are untested in practical scenarios.
    - No code or synthetic data generation scripts provided. Results use single runs with no error bars or statistical tests.
    - OOD definitions (e.g. longer sequences, more steps) are synthetic, not true distributional shifts as encountered in real domains.
3. **Lack of Novelty Relative to Recent Work**
    - The key claim that GRPO is a conservative reweighting mechanism bounded by pretraining is already well established [1, 2, 3].
    - The contribution is mainly the synthetic experimental isolation and most theoretical and empirical results duplicate known findings.

### **References**

[1] Samineni S. R. et al. *RL in Name Only? Analyzing the Structural Assumptions in RL Post-training for LLMs.*

[2] Wu Y. et al. *On the Generalization of SFT: A Reinforcement Learning Perspective with Reward Rectification.*

[3] Zhao R. et al. *Echo Chamber: RL Post-training Amplifies Behaviors Learned in Pretraining.*

**Questions:**

Could you please address the concerns in the weaknesses list?

---

### Meta-Review · Area_Chair_gLx6 · 2025-12-10

**Summary:**

The common issues lied in the theoretical limitations, experimental design weaknesses, and lack of novelty relative to recent work.

The authors did not reply to the review comments.

It has to be rejected.

**Reviewer Concerns:**

No rebuttals.

**Reviewer Scores:**

It received 4 reviews with the average score 2.5.

No discussions for no rebuttals.

---

### Decision · Program_Chairs · 2026-01-26

Reject